# Scoping Challenges and Opportunities Presented by COVID-19 for the Development of Sustainable Short Food Supply Chains

Nuno Baptista [1,2,*], Helena Alves [2] and Nelson Matos [3]

1    Escola Superior de Comunicação Social, Instituto Politécnico de Lisboa, Campus de Benfica do IPL,
     1549-014 Lisboa, Portugal
2    NECE-UBI, Universidade da Beira Interior, Convento de Sto. António, 6201-001 Covilhã, Portugal
3    CINTURS, Universidade do Algarve, Estrada da Penha, 8005-139 Faro, Portugal
*    Correspondence: nbaptista@escs.ipl.pt; Tel.: +351-217119000

**Abstract:** Over the past decades, short food supply chains attracted government and public support owing to their potential to mitigate some of the sustainability issues associated with the conventional globalized food supply system. The recent event of the coronavirus disease pandemic placed unprecedented pressure on food supply systems worldwide, and it constitutes a unique opportunity to evaluate the performance of food chains. Through a scoping review of the academic literature, this study provides a critical assessment of the implications of the pandemic on short food supply chains in multiple economies. Following the guidelines outlined in the PRISMA-ScR framework, the SCOPUS and ISI Web of Science databases were searched for the academic literature on the topic. The results of the review indicate that, besides the direct effects of the pandemic, the indirect effects resulting from public policies implemented to contain the spread of the virus affected all relevant dimensions of sustainability. Moreover, the consequences of the pandemic were more disruptive in the short food chains of low-income countries than in those of high-income countries. The main challenges and opportunities for the sustainable development of short food supply chains are identified, and recommendations for future research are outlined.

**Keywords:** sustainability; short supply chains; food; pandemic; COVID-19

## 1. Introduction and Background

During the 20th century, food markets transformed significantly under the impacts of globalization, increasing urbanization, liberalization of trade, technological changes, resource scarcity combined with the growing world population, and shifts in consumption patterns [1–3]. The current dominant system, based on industrialized production and globalized food delivery, successfully provided a cheap and diverse supply of food to a growing population, including in low-income countries [4]. However, the dominance of multinational corporations, global integration, standardized organization, long-distance transportation, long supply chains, and the mass production features of the current dominant food supply system raised widespread concerns about its social, economic, and environmental sustainability [1,5]. At the center of these concerns are the pollution of soil and water, greenhouse gas (GHG) emissions, excessive land use, extensive food waste, loss of biodiversity, unfair distribution of the economic value created among supply chain members, poor working conditions for agricultural workers, and adverse impacts of food on human health [1,2,5,6]. Moreover, consumers are becoming increasingly concerned about the traceability, quality, and safety of food products [7], especially as conventional food supply chain systems suffer from a confidence crisis and have weak affiliated values [5].

Amidst the discontent and criticism surrounding conventional industrialized food supply systems owing to their potential to mitigate some of the sustainability issues associated with these systems, alternative food production and distribution schemes garnered interest in

academia and policy-making circles in the last 20 years, including organizational models based on locally grown and distributed food involving short food supply chains (SFSCs) [8–10].

The most intuitive and commonly cited feature of SFSCs is that they involve some form of network, throughout which food products move from the production to the consumer point and where the number of intermediaries is reduced [11]. However, SFSCs are a multifaceted concept and there is no consensus on a unique and universally accepted definition [12,13]. In this article, we adopt the definition articulated by the European Parliament and of the Council in regulation No 1305/2013 [14], that reflects the previously mentioned characteristics, and defines SFSC as a supply chain that has a limited number of economic operators, is committed to co-operation, local economic development, and involves close geographical and social relations between producers, processors, and consumers. Consistent with this definition, SFSCs can assume various forms, including, for example, farmers' markets and fairs, farm shops, box delivery schemes, pick-your-own models, community-supported agriculture, consumer cooperatives, internet sales, and farmers' direct sales to small retailers [1,15,16].

The outbreak of the coronavirus disease 2019 (COVID-19) and the consequent implementation of stringent measures by world governments in an effort to isolate cases and limit the transmission of the virus disrupted the global food supply chain, thereby exposing the vulnerabilities and revealing that some parts of the system are not resilient to disruptions outside the normal range [11,14,17,18]. Considering the vulnerability of the global food supply system exposed by COVID-19, to strengthen the resilience of the system, some authors argue for a multichannel approach to food supply and propose complementing the dominant system with local food production and short supply chains [12,14,15,17,18]. A common assumption in some of the research literature, as well as in political discourse, is that compared to conventional industrialized food supply systems, SFSCs are more economically, socially, and environmentally sustainable [19–22]. However, to date, empirical evidence supporting the supposed sustainability of SFSCs is scarce, and this proposition needs further exploration [3,14,19,20].

The sustainability of supply chains is one of the most explored topics in supply chain management [13,23]. Since Spreckley (1981) [23] articulated the "triple bottom line" framework, which attracted academic and public interest after the work of Elkington (1997) [16], sustainability in supply chains began to be interpreted in terms of the three dimensions of economic prosperity, environmental quality, and social justice. More recently, Dos Santos and Ahmad (2020) articulated an institutional dimension, which refers to the country's level of institutional support and policies that directly or indirectly promote sustainability [24].

This article focuses on the sustainability of SFSCs and attempts to answer the following research question: what are the observed and potential effects of the COVID-19 pandemic on SFSCs in terms of institutional, economic, environmental, and social sustainability? The main objective of this research is to uncover the challenges and opportunities presented by COVID-19 for the future development of sustainable SFSCs. The COVID-19 pandemic constitutes a unique opportunity to evaluate the sustainability performance of food chains under unique stress conditions. In general, research on the impact of pandemics on the food supply system is limited [25]. Notwithstanding the growing literature on the impact of the COVID-19 pandemic on specific food markets, short food systems, and countries, this topic is not yet systematically explored.

The ongoing situation of the pandemic necessitates a research approach that is exploratory in nature and can systematically search, select, and synthesize existing knowledge. Through a scoping review of the academic literature, this article examines how SFSCs are performing during the pandemic, while assessing the observed impacts of COVID-19 on SFSCs from a sustainability perspective across countries and food systems. Considering the need to transition toward more resilient food supply chains that meet the United Nations (UN) Sustainable Development Goals (SDGs), the findings of the present research systematize existing knowledge and provide insights that can be useful in informing policy discussions and orienting future research.

## 2. Materials and Methods

A scoping review of the academic literature was undertaken to attain the objectives of the study. Scoping reviews are a specific method of knowledge synthesis that can be applied to present a broad overview of a research topic by addressing exploratory research questions aimed at mapping gaps in research related to an emerging field and setting research agendas [26–28]. As highlighted by Tricco et al. (2016) [29], scoping reviews differ from systematic literature reviews in the sense that while the former is exploratory in nature, being commonly used to examine new areas that are emerging, the latter is more suitable for addressing specific questions related to more mature research topics. Although conducted for different purposes compared to systematic reviews, scoping reviews also require rigorous procedures to ensure that the results are trustworthy [29,30]. Considering the exploratory nature of the research, the authors opted to conduct a scoping review instead of a systematic review. The review process employs a systematic approach based on the PRISMA Extension for Scoping Reviews (PRISMA-ScR) from Tricco et al. (2018) [31] (Appendix A). The research project and corresponding protocol are available in the Open Science Framework (OSF) database, and can be downloaded at https://osf.io/dq3rt/?view_only=afca63ea0a604abb901995618192b3ef (Supplementary Materials).

The three authors searched the ISI Web of Science and Scopus databases for academic articles on the topic published up to 25 September 2021. Non-English articles, as well as editorials and conference proceedings were excluded. The screening process entailed the following search terms in the title, abstract, and keyword fields: "short food supply chain"; "alternative food network"; "local food*"; "sustainab*"; "economy"; "social"; "triple bottom line"; "environment"; "COVID-19"; "SARS*"; "pandemic"; and "lockdown". The Mendeley software was used to store the original articles and remove duplicated records.

The retrieved articles were independently screened for relevance by all authors. The screening and selection of articles were performed in two phases: the first stage covered the titles and abstracts, and the second stage focused on full texts. At both screening stages, the inter-rater reliability, based on percentage agreement, was computed. A considerable level of agreement was observed in both phases of the process (>80%). Based on the reading of the articles, a snowballing technique was applied to identify other articles relevant to the investigation. Opinion divergences regarding inclusion criteria were resolved through discussion, leading to a final sample of 46 articles for subsequent content analysis. The screening process is illustrated in Figure 1.

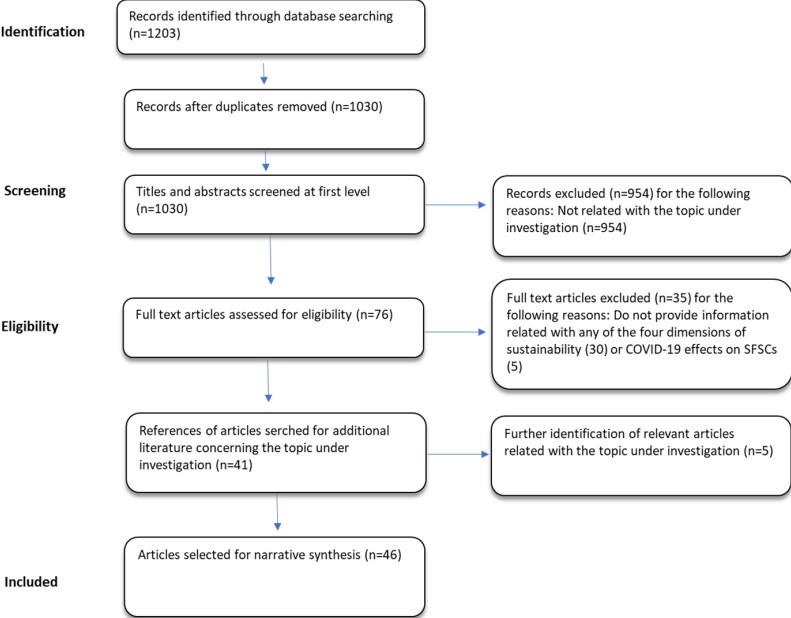

**Figure 1.** PRISMA flowchart.

The authors independently coded the selected final sample of the articles. The coding scheme, which can be consulted in the study's protocol, was partially based on the short supply chain knowledge and innovation network (SKIN project) as operationalized by Jarzebowski et al. (2020) [12], and reflects the four pillars of sustainability (economic, social, environmental, and institutional). The inter-rater agreement was high between the researchers, reaching a Cohen's kappa coefficient of $\kappa = 0.74$, implying substantial agreement [32,33]. Discrepancies in codification were solved through intense discussion among the authors and ultimately agreed upon.

## 3. Results

As detailed in Table 1, the majority of selected articles adopt a global perspective by focusing on multiple regions of the world (n = 17; 37%). Some cover European countries (n = 8; 17%), such as Italy, Greece, Italy, Portugal, and Western Balkan countries; other papers focus on South Asian countries (n = 7; 15%), most predominantly India; some analyze North American countries (n = 5; 11%), including the US and Canada; some center on East Asia and Pacific countries (n = 5, 11%), especially China; and finally, some articles focus on African countries (n = 4; 9%), including South Africa, Ethiopia, Nigeria, and Senegal. In terms of research methods, the authors privilege case studies (n = 24; 52%) and theoretical/conceptual studies (n = 14, 30%). Moreover, most articles involve qualitative methodologies (n = 30; 65%), and a smaller number resort to mixed quantitative/qualitative methods (n = 8; 17%) and quantitative methods (n = 8; 17%). The most represented journals are *Sustainability* (n = 8; 17%) and *Agricultural Economics* (n = 8; 17%). In terms of topics covered, most papers focus on the economic sustainability of SFSCs (n = 31, 67%) and the institutional dimension (n = 27, 59%), while the environmental (n = 12, 26%) and social dimensions (n = 14, 30%) are less explored in the reviewed literature.

**Table 1.** Overview of the articles included in the scoping review.

| Geographical Reach | Methodology | Year | Journal |
|---|---|---|---|
| Global (17) | Case study (24) | 2021 (33) | *Sustainability* (8) |
| Europe (8) | Theoretical (14) | 2020 (13) | *Agricultural Economics* (8) |
| South Asia (7) | Literature Review (6) | | *Trends in Food Science & Technology* (3) |
| North America (5) | Simulation/modeling (2) | | *American Journal of Agricultural Economics* (2) |
| East Asia and Pacific (5) | | | *Global Food Security* (2) |
| | | | *International Journal of Environmental Research and Public Health* (2) |
| Sub-Saharan Africa (4) | | | *Journal of Food Science and Technology* (2) |
| | | | *Resources, Conservation & Recycling* (2) |
| | | | *Science of the Total Environment* (2) |
| | | | Others (15) |

Compared with traditional food systems, SFSCs are supposed to offer a fairer distribution of value among chain actors (e.g., intermediaries' fees are reduced and selling price is more controllable) and involve a higher level of trust, transparency, cooperation, and shared governance between supply chain participants, thus decreasing economic and relationship uncertainty [13,15,34,35]. From a consumer perspective, SFSCs offer food with superior nutritional and health values, facilitate the identification of the food place of origin, and promote direct contact between the consumer and the producer, a facet that many consumers increasingly value [1,3,36]. However, the economic sustainability of SFSCs is questioned by some authors. Most arguments are centered on the limited demand for the nature of products typically offered in these chains (remains a market niche) coupled with the frequent misalignment of consumers' sustainability intentions and actual purchasing

behavior (attitude–behavior, intention–behavior, or words–deeds gap), the premium prices usually practiced, higher unitary production costs, and the general difficulty of small producers operating in SFSCs to secure a long-term stable position in the food market [34]. From a social perspective, SFSCs contribute to food quality and security, are a factor of local job creation, foster social inclusion, promote gender equality (women often participate in rural work), and contribute to preserving cultural heritage [1,15]. Furthermore, SFSCs are expected to enhance local development and boost the local economy by increasing local financial flows and supporting synergies with other sectors, such as tourism [6,15,36–38]. The importance of the institutional dimension of sustainability is highlighted in the Sustainability Assessment of Food and Agricultural Systems framework from the Food and Agriculture Organization of the UN that equally considers a "rule of law" theme under a "good governance" dimension of sustainability [39]. The relevance of short chains in achieving sustainable development is reflected in the UN 2030 Agenda for Sustainable Development, which stresses the importance of family farming, local production, and SFSCs as an effective way to tackle poverty across the globe [26].

The COVID-19 pandemic revealed the lack of resilience of the dominant food distribution system in responding to unplanned shocks and crises in the short term, thus exposing the weakness of complex interdependent globalized supply and production chain networks [40,41]. The difficulties faced by the conventional food supply system in the initial stage of the pandemic prompt academics to question whether SFSCs are in a better position to adapt to the crisis [17,18,39,42,43]. Local producers in SFSCs are usually more integrated with customers and downstream supply chain actors in short circuits because they focus on fulfilling local markets, tend to be less dependent on export inputs, and are theoretically more resilient to market shocks and demand volatilities [44]. Table 2 summarizes the effects of the pandemic in some selected economies based on the reviewed literature.

**Table 2.** Effects of the COVID-19 pandemic in some selected economies.

| Country | Category | Findings | References |
|---|---|---|---|
| United States | High income | Increased consumer demand for regionally produced food with momentary price spikes for most farm commodities. Shortages of some fruits and vegetables. Bulk ocean transportation disturbed. Incidents of perishable products being disposed. Difficulties to obtain adequate labor. Increased role of online shopping. Consumers reduced the frequency of food shopping. | [40,45–51] |
| China | Upper middle income | Expansion of local food production models, such as Community Supported Agriculture in peri-urban areas. Shortages of inputs, such as feed, animal vaccines, and fertilizers. Consumers forming habits of ordering fresh food online. Farmers recurred to e-commerce platforms to sell produce. | [11,21,52] |
| India | Lower middle income | Intermittent closure of wet markets for fish and meats, as well as weekly farmers' markets. Street vendors of fresh produce forced to suspend their business. Severe supply bottlenecks. Unavailability of raw materials required for food production. Panic-buying. | [17,48,50,53–55] |

**Table 2.** *Cont.*

| Country | Category | Findings | References |
|---|---|---|---|
| France | High income | Logistic and transport limitations. Within a few months, the system managed to fully recover from the initial crisis. Consumers positively changed their views on the social, economic, and ecological value of food production during the quarantine. | [45,47] |
| Romania | Upper middle income | Consumer interest increased for local, fresh, and highly nutritious food. Change in purchase behavior toward direct delivery of fresh vegetables from local producers within the quarantined areas. Preference for reduced time periods of ordering. Growing confidence in the consumer–local producer relation. | [31] |
| Senegal | Lower middle income | Local producers were affected by mobility restrictions, closure of wet markets, and lack of financing and cold chain infrastructure. SFSCs showed little capacity to adjust and innovate in response to the shock. Increased demand for online commerce. | [56,57] |
| Ethiopia | Low income | Marketing margins declined for several vegetable products. Inter-regional trade restrictions led to a more localized marketing system. Shifts in the pattern of food consumption toward staples and away from vegetables and legumes. Widespread myth that eating raw vegetables would increase the likelihood of contracting the virus. | [57,58] |

Our analysis indicates that the pandemic affected all the components and activities of local food production of SFSCs, with relevant implications in terms of the sustainability of the systems. These results will be further explored in the next sub-sections that analyze the impacts of the pandemic on SFSCs from the perspective of the economic, social, environmental, and institutional dimensions of sustainability.

*3.1. Economic Sustainability*

The COVID-19 pandemic is not only a global health crisis, but also an economic one, wherein the effects are being felt across many countries and sectors of the economy, including the food sector, and SFSCs in particular [40,41,59]. On the supply side, short circuits and small farms in most countries faced difficulties in accessing agricultural inputs, disruptions caused by labor shortages, and logistic bottlenecks, and were strongly affected by the closure of open-air markets [31,46,56,60]. Local production and SFSCs were affected by movement restrictions because local agricultural products that could not be transported out during the enforcement of movement restriction by some governments often resulted in food loss and waste, especially perishable products, thereby leading to unsold agricultural products and the consequent loss of income for small farmers [46,61]. The closure of open-air markets in numerous countries prevented local producers from selling their products directly to consumers and accessing necessary inputs and extension services, with rural farmers reportedly experiencing marginalization and negative discrimination in relation to supermarkets [62–64]. Small farmers who were able to migrate to online selling faced high demand during the lockdowns and struggled with excessive orderings, thus compromising in-time deliveries and customer satisfaction [33].

In Europe, small farmers faced some logistic difficulties due to movement restrictions and a shortage of seasonal immigrant workers, particularly in the fruit and vegetable sectors [4]. Notwithstanding the absence of significant changes in food supply, prices of meat products fell during the lockdowns and shortages of some vegetables slightly increased in price in European countries [4]. The crisis reinforced some pre-existing trends, with an increase in demand for locally produced food and e-commerce sales, while nutritional quality, origin, and health concerns became more prominent in consumers' choices (EU, 2020). In a survey conducted by Coopmans et al. (2021) in Flanders, Belgium, farmers selling through SFSCs reported fewer negative economic effects from the pandemic than farmers selling all of their produce in wholesale markets [65]. For example, approximately 41% of farmers selling through SFSCs reported lower prices compared with 80% of farmers selling in wholesale markets [65]. In France, most farmers involved in organic dairy cattle farming reported minimal impacts of the pandemic, as farmers were able to shorten their supply chain and deliver milk and dairy products directly to supermarkets, thus avoiding sub-level platforms and other intermediaries [41]. The exponential increase in online deliveries constitutes a relevant impact from the pandemic in most European states [45], as well as the dramatic repercussions of the crisis in the agritourism sector associated with SFSCs, such that the demand reduced significantly [66].

In the developed economies of North America, such as the USA and Canada, the effects of COVID-19 on food availability and prices were limited [47,48]. Notwithstanding some food price momentary increases and supply chain disruptions, there were largely no widespread declines in affordable food as a consequence of the pandemic [27]. In the USA, farmers faced reported difficulties in obtaining labor because the pandemic limited travel and restricted the movement of foreign workers that were essential to the production of several farm products, such as fruits and vegetables [27], and farm-to-table restaurant demand disappeared with the restrictions imposed on the HoReCa channel [46]. Employee illness, closed borders, and mobility barriers imposed to inhibit the spread of the virus, reduced the workforce supply necessary for harvest and collection, thereby resulting in a lack of seasonal workers in the fruit and vegetable sector [50,63]. This circumstance affected small local firms because, despite the use of family labor for production, they are often more manual labor intensive than large operations, and thus are more vulnerable to workforce supply disruptions [33]. Movement restrictions and border closures also negatively impacted the delivery of essential agricultural inputs, such as fertilizers, seeds, or pesticides for many small farmers in SFSCs, thereby spiking prices [33]. The demand for locally produced fruits increased due to the restrictive measures on importation and the need felt by consumers to strengthen the immune system [60]. However, the pandemic reduced price premiums for credence attributes in some products, such as vegetarian-fed and organic eggs, by as much as 34%, and prices did not fully recover following the normalization of the economic activity [56]. With the closure of the HoReCa channel, some small-scale local producers were able to rapidly pivot to sell directly to consumers by increasing their staff and acquiring delivery vehicles to provide contact-free home deliveries [46]. In addition to pivoting strategies for e-commerce, small farmers were able to resist the crisis by changing delivery intermediaries and reinforcing mutual assistance among farmers [47,57,60]. In the USA, farmers also reportedly suffered from increased costs due to anti-pandemic measures, including the implementation of stricter security measures and regulations, such as labeling and packaging of products, contactless distribution, high consumption of disinfectants, and an increase in the price of agricultural inputs [18,67].

In economic-developing Asia, the pandemic impacted smallholders, who are still a major part of the food system, with 90% of food processors and manufacturers being small and medium-sized enterprises (SMEs) [54]. The slowing economy caused widespread job loss, collapsing incomes, and falling remittances in the region [68]. In India, an agrarian country with a long history of traditional food processing practices and SFSCs, wherein the food market is highly dependent on SMEs [17,53], the lockdown involved intermittent closures of wholesale agricultural markets, wet markets for fish and meat, and restrictions

on the movement of vehicles, both across state borders and within cities [63]. The country lockdown coincided with the peak of harvesting time for various crops, such as summer vegetables and fruits. Thus, the lack of harvesting labor caused huge food waste and economic losses for farmers [55]. These restrictions in India brought forward the vulnerabilities of SFSCs under lockdown, with rural farmers reportedly experiencing marginalization [51]. However, the consequences of the pandemic for long chains appeared to be more severe. Mahajan and Tomar (2020) [64] computed the distance to production zones from retail points and found that the fall in product availability in India was larger for items cultivated or manufactured further from the point of sale, inferring that long-distance food supply chains were hit the hardest in India during the current pandemic.

In China, the lockdown measures and mobility restrictions affected the transport of agricultural inputs, and labor was in shortage [69]. In response to labor shortages, farmers employed returning migrant workers from the cities and developed a mutual aid system within villages to auxiliate harvesting and ensure the supply of agricultural inputs needed for production [69]. The decline in sales at wet markets due to government closures of these retail outlets was followed by a significant increase in online food purchases and interest in community forms of organizations. Large e-commerce firms, such as Alibaba and Pinduoduo, supplanted the traditional intermediation between farmers and consumers, involving brokers, wholesalers, and brick-and-mortar retailers, and established direct linkages between small producers and consumers [49]. The pandemic also led to an increasing interest of consumers in local organic food production models, such as community-supported agriculture in peri-urban areas [61]. Overall, food prices in China remained relatively stable, and the food supply of fruits and vegetables was able to meet demand, despite sporadic reports of price hikes and shortages in more isolated locations [69].

Notably, SFSCs in developing economies were more affected by the COVID-19 pandemic than in developed economies [57]. The loss of jobs and consequent reduction in household income affected consumers' purchasing power and reduced demand for agricultural products in African countries, where small-scale farmers produce most of the food consumed [52]. However, panic food purchases, as observed in countries such as Rwanda, South Africa, Kenya, and Nigeria, caused local food prices to increase [52]. The unavailability of raw materials and the reduction in labor availability of up to 25% were also reported as potential factors affecting the sector in Africa [60]. For example, in Nigeria, the lockdown imposed on wet retail markets across the country disrupted the operations of food companies in SFSCs, and movement restrictions deprived companies of supporting logistics, agricultural inputs, finance, materials, and labor [53]. In Senegal, a study based on a phone survey to compare the resilience of vertically integrated companies and the more traditional domestic-oriented supply chain in the horticulture sector during the pandemic revealed that during the first months of the pandemic, large integrated businesses in the food supply sector were less severely affected than SFSC companies [70]. According to the authors, large integrated companies in Senegal were able to adapt by optimizing their workforce (double shifts and protective gear) and investing in safer transport vehicles. In contrast, small companies in SFSCs were severely affected by mobility limitations and the closure of wet markets due to the low level of organization, and they showed limited capacity to adjust and innovate in response to the crisis.

### 3.2. Social Sustainability

In the context of food supply chains, the social sustainability dimension refers to the contribution of the system to trust and fairness among chain actors, as well as the lack of prejudice and social exclusion, which are rooted in notions of solidarity, social cohesion, and social well-being [3,6]. In response to the COVID-19 pandemic, there was an increase in popular sentiment toward local food production and SFSCs as a way of supporting local communities [4,7]. SFSCs can strengthen connections between producers and consumers, sustain small farm businesses, and allow other intangible gains, such as the reinforcement of the sense of community, cultural bonds, collective values, and traditions [14,15,71,72].

The reinforcement of local economies through SFSCs is particularly useful in a period of crisis, as it supports local economic regeneration and improves citizens' well-being through job creation, thus constituting an instrument to support local communities [6,73,74]. In addition, income generated by SFSCs may remain in the local economy, generating additional taxes, improving territorial governance, and financing other local activities and investments [34].

From a social perspective, the most significant negative effects of SFSC disruptions during the pandemic were growing unemployment and increased food insecurity. In several African countries, family farmers and other small-scale food producers in the food sector accumulated losses, which resulted in increased sector unemployment [75]. In low-income countries, farmers and workers tend to be younger than in high-income countries, the health systems are usually deficient, and significant health challenges are presented to small businesses in SFSCs to secure labor [47]. Research indicates that female employment was particularly affected by COVID-19 restrictions. For example, according to a survey in Nigeria, between February and April 2021, the share of fish businesses employing women in the fish industry plummeted from 20% to as low as 2% in April [53]. In Africa, women are traditionally involved in open market trade, and the enforced closure of these outlets affected women's employment and income. Moreover, on a global scale, COVID-19 affected women's work due to school closures and the need to care for children and sick household members [57,67].

In the developing regions of Africa and Asia, micro, small, and medium-sized enterprises in SFSCs play an important role in the agri-food system to ensure the achievement of the SDGs relating to food and nutrition security [60]. COVID-19 severely affected SFSCs in these countries, where the population is particularly vulnerable to poverty, hunger, and malnutrition. The pandemic affected all dimensions of food security, including availability, access, use, and stability, because of the unpredictability of food prices, food supply disruptions, reduced dietary diversity, lack of labor for planting and harvesting, consumers' panic purchases, and the negative impacts of lockdowns on families' incomes [18,46,76].

Food security in some developed economies was also affected by momentary increases in food prices in most countries [76] and reported difficulties faced by food provisioning services to serve vulnerable populations in the context of a growing number of citizens resorting to food banks and other sources of social assistance [4]. For example, in the USA, the National School Lunch Program disrupted services to the juvenile population due to school closures [75], and in the UK, people from ethnic minorities and with existing health conditions were most at risk, owing to the government's weak response to the crisis and the lack of additional social protection measures [77].

Curiously, while strong community and cultural values are often considered to favor the support for local production and SFSCs [6,14], in some instances, these can be counterproductive. Strong cultural values provide shared representations, interpretations, and systems of meaning among parties, such as accepted narratives [78]. During the early days of the COVID-19 pandemic in Nepal, there was a popular myth that the virus was transmitted through food products, including fresh vegetables, which caused a decline in the demand for these agricultural products [57]. Similarly, in Ethiopia, a reduction in the demand for vegetables was associated with the widespread fear that eating raw vegetables would increase the likelihood of contracting COVID-19 [47]. These cases exemplify how trust based on cultural attachment can override cognitive weighing and rational thinking in a pandemic context.

In Europe, the pandemic raised public awareness about the hard-working and living conditions of those on informal or seasonal rural employment and migrant workers in the fruit and vegetable sector. In response, Germany promised to make use of its European Union (EU) presidency to push for stricter enforcement of work protection legislation, and the EU Commission published guidelines highlighting the importance of improving immigrant workers' conditions, including remuneration, dismissal, and occupational safety and health protection [4,40]. Unemployment among food sector workers in the EU increased significantly during the pandemic [75]. Unemployment affected remittance transfers by migrant workers to their families back home, thereby impacting household

income in developing countries [47]. As many immigrants are informal workers, they do not benefit from welfare systems or unemployment compensation schemes [21,48].

*3.3. Environmental Sustainability*

The food production system contributes to approximately one-third of the total GHG emissions derived from human activity and is thus considered a main factor in climate change and ecosystem degradation [9]. The measures implemented by governments worldwide to reduce the spread of COVID-19 led to a reduction in ecosystem carbonization, owing to reduced traffic, air travel, and energy consumption [11,18,50]. Nonetheless, the impact of food system adaptation to the pandemic on global GHG emissions was not very significant. Elleby et al. (2020) estimated the global impact to be $-0.2\%$ in 2020 and $-1.1\%$ in 2021, and projected $-1.0\%$ in 2022 [79]. The authors estimated a larger reduction for some large producers, such as China and the USA, amounting to $-2.3\%$ in 2022 [79]. Moreover, post-pandemic economic recovery may force countries to embrace policies favoring the improvement of the industrial complex and high-carbon technologies, capable of projecting carbon emissions back and even above pre-pandemic levels [60]. Modeling projections from the EU indicate that carbon GHG emissions from EU agriculture would remain largely unchanged because of the increase in nitrous oxide emissions due to higher crop yields [67]. Moreover, the verified reduction in pollution linked to the contraction of the economy may obscure the perception of climatic urgency and lead governments to prioritize other investments [66].

In some studies, SFSCs are assumed to be less harmful to the environment in terms of fossil fuel energy consumption, pollution, and GHG emissions because of the usual shortened physical distance between producers and final consumers, requiring less transportation, cold storage, processing, and packaging compared with conventional long supply chains [3,49]. More environmentally sustainable production and processing methods tend to be associated with SFSCs, resulting from less use of pesticides, synthetic fertilizers, animal feed, water, and energy consumption [34]. As noted in the background section of the present article, several studies suggest that SFSCs are beneficial to the environment, albeit they do not provide substantive evidence to generalize this claim [3,5,14,20].

In a recent study conducted by Majewski et al. (2020) [3] comparing the environmental impacts of several typologies of SFSCs and long chains, based on various eco-efficiency indicators (global warming potential, acidification, eutrophication, ozone depletion, photochemical potential, ozone emission, and non-hazardous waste disposed), the authors found that, on average, SFSCs are less eco-efficient than long chain models. Short chains do not necessarily entail a more efficient delivery system. In addition to the reduced number of intermediaries, geographical proximity, and social proximity, which are the main characteristics of SFSCs, other equally important intrinsic and contextual factors affect the environmental efficiency of the chain, including, for example, the diversity of food products, the impact of the product life cycle and production methods, logistics organization and technology, the specific characteristics of each consumer market, the diverse modes of transportation, equipment, and types of fuel, all of which make generalization particularly difficult [6,12].

The effects of the COVID-19 pandemic on the environmental performance of SFSCs are not explored in the literature thus far. The growing adherence to e-commerce by small farmers and businesses in the food chain can potentially generate positive environmental externalities, as specialized e-commerce platforms usually involve optimized logistics and improved organization. The literature describes how small producers resort to major e-commerce platforms, such as Pinduoduo, Alibaba, or Jumia, to sell their products, with these major companies assuming the delivery function [56,60]. For example, during the pandemic, Alibaba developed a specialized e-commerce platform to assist Chinese farmers in selling their unsold agricultural products, and is also developing a "green channel" for fresh food products [19]. This trend may be reinforced in the future as big players, such as Amazon, who recently acquired Whole Foods, are showing the intention to enter the fresh food market [27].

*3.4. Institutional Sustainability*

The main policies implemented at a global scale to reduce the spread of the virus involved citizens' confinement, restrictions on movement, international commerce trade restrictions, as well as the closure of local food and peasant markets, schools, social assistance centers, and food shelters. These measures resulted in significant pressure over different functions and actors that are crucial for the appropriate functioning of SFSCs, as previously described. However, governments also implemented supporting policies targeting local producers and SFSCs.

Governments' policies to support local SFSCs producers to cope with the pandemic reflected differences in natural resources, organization of the food market, and socio-economic conditions [28]. Reactive measures adopted by governments to reduce the impact of the pandemic on SFSCs mostly included: (i) Accessing new markets—policy measures to support smallholders in accessing online markets and developing e-commerce. (ii) Production support—measures to improve productivity, reduce food waste, and management and destruction of production surplus. (iii) Employment policies—policy measures to secure agricultural labor, including allowing the free movement of agricultural workers, combining work with unemployment subsidies and allowing the border entrance of seasonal workers without work permits. (iv) Distribution support—measures to facilitate the distribution of food products, such as allowing the free movement of food products, subsidies to improve supply logistics, measures to support digital payments, and cold storage support. (v) Price regulation and support to disadvantaged groups of consumers—policy measures to regulate food commodities prices and combat speculation, as well as food support programs targeting specific segments of consumers. (vi) Financial support—safety net programs for individuals running small businesses, public procurement (e.g., food baskets), exemption of social security payments, and extended deadlines to state payments [33]. Next, we highlight some regional measures adopted by countries in specific regions of the world.

In the EU, the impact of supporting policies was undermined by the limited funds allocated. The European Commission developed some measures to support farmers' income during the lockdowns. However, the Commission's response was framed by the limited resources available in the EU budget during the final year of the budget for 2014–2020, revealing the limitations of the EU's crisis response mechanism, when support for small farmers was most needed [4]. The Commission's direct response involved greater flexibility in the rules governing the disbursement of common agricultural policy payments, as well as temporary derogation from EU competition rules for producers and other forms of direct aid [4]. By making EU rules more flexible, the EU transferred responsibility to member states, which provided direct financial support to farmers, with the largest national support schemes being approved in Italy, the Netherlands, Hungary, and Czechia [4]. In the USA, the U.S. Department of Agriculture (USDA) supported small farmers through pandemic response and safety grant programs that covered diverse expenses, such as workplace safety measures, retrofitting facilities, shifting to online commerce, transportation, and medical costs [80]. The USDA also introduced special financial programs that supplemented marginal coverage for small and medium dairy farmers and allocated cost share assistance to organic producers and producers transitioning to organic [80].

In Asia, a wide array of state and government policies to support small producers were implemented. China, India, Japan, South Korea, Singapore, Indonesia, Thailand, Malaysia, the Philippines, and Vietnam provided government loans, loan guarantees, tax breaks, and subsidies for enterprises in the food and agriculture sectors [81]. These countries, with the exception of India, South Korea, and Indonesia, also provided employment subsidies and lockdown exemptions for food chain workers [81]. In Cambodia, India, and Fiji, government policies to support local farmers included measures targeting job creation, education on agricultural techniques, and tax incentives for migrant workers [60]. In China, the central government established subsidies for purchasing machines and tools for agricultural purposes, provided low-interest rate loans, and developed specific programs

supporting cold chain storage and logistics to facilitate online commerce of agricultural products [11,46,52]. Finally, in developing Africa, where major restrictions were placed on food markets, government support was mostly directed to maintain consumer livelihoods. Although some support programs were implemented, the agriculture sector, and in particular, smallholder farmers, received less economic assistance than other sectors [54].

## 4. Discussion

Overall, the research findings suggest that the effects of COVID-19 on SFSCs are context-specific and dependent on the features of each market and the type of policies adopted by governments in response to the pandemic.

Even though the health crisis is still ongoing, it is possible to infer that in general, the pandemic's effects on SFSCs in the food sector were more disruptive in low-income countries than in high-income countries. It is also possible to conclude that some policy measures to avoid infection spikes, such as restrictions on movement among regions, the closure of farmers' markets, and disruption in transportation systems of people, goods, and inputs, significantly affected the functionality of SFSCs, with negative impacts on sustainability. In low-income countries, SFSCs appear to be more affected than capital-intensive modern food chains, while in high-income countries, both long integrated food chains and SFSCs were able to more rapidly recover from the crisis. The rapid recovery of SFSCs in developed economies resulted from the increased demand for nutritional quality food, and companies' ability to adapt to a new context by rapidly developing an online presence or relationships with e-intermediaries, investing in transportation facilities, and diversifying their supply channels [82]. The shift toward the online world was manifested in many different forms, from e-procurement to online advertising via social media platforms (Facebook, Reddit, and WhatsApp groups), proprietary online shops, and partnerships with established international e-commerce platforms [70], and this trend is likely to persist after the pandemic.

On the demand side, the pandemic increased consumer awareness about the importance of healthy diets and food safety. In general, the literature indicates that the popularity of local food and SFSCs was positively impacted as a direct consequence of restrictive measures on the importation of food and the desire of consumers to follow a healthy diet to protect themselves and their immune systems against the COVID-19 contagion [45,50,74]. Through SFSCs, people could experience the quality of healthy local food and avoid perceived contamination risks in agglomerations at supermarkets, and in many locations, including developed and underdeveloped economies, local food producers were able to start selling through online channels and implement direct-to-your-door delivery systems that allowed consumers to access fresh produce more safely during the lockdowns [7,60]. Research also revealed an inclination for consumers to increase the frequency of purchasing and buying according to their momentary needs rather than buying pre-defined baskets [31,53].

Consumers are forming habits of ordering fresh food online, and big e-commerce platforms facilitate the supply of fresh agricultural production. This tendency, which was already visible before the pandemic in developed regions, spread quickly to emerging and developing regions [47]. Online sales channels constitute a relevant opportunity for the future development of SFSCs. Through online commerce, consumers can more easily compare alternatives and obtain complementary information online about food product offers, thus contributing to transparency in the chain. The reduction in intermediaries proportionated by e-commerce can also contribute to a fairer distribution of incomes and value appropriation in the food supply system and can potentially strengthen relations between producers and consumers toward more trusting, equitable, and fair commerce.

### 4.1. Sustainability Impacts of the COVID-19 Pandemic

The multiple dimensions of sustainability are not always complementary. The lockdowns and consequent movement restrictions, as well as the closure of open markets, both of which caused serious difficulties for SFSCs, potentially produced positive marginal

effects on environmental sustainability and stimulated some level of technology adoption by SFSC producers; however, they implicated a series of negative impacts on SDGs related to social sustainability, including SDG 1 (no poverty) SDG2 (zero hunger), SDG3 (good health and well-being), SD5 (gender equality), SDG 8 (decent work and economic growth), and SDG10 (reduced inequalities) and SDG11 (sustainable cities and communities). The pandemic crisis affected businesses in SFSCs, causing cash flow shortages and diminishing operating capacities with negative effects on employment and small producers' income.

There may be a trade-off between supply chain sustainability and resilience. The small-scale operations, non-intensive production systems, and the use of organic inputs, which are often the argumentation basis for SFSC sustainability advantages, limit the capacity of SFSCs to constitute a real alternative to the conventional supply system. The pandemic underlined the importance of a resilient food supply system that functions during extreme events and is able to rapidly adjust to market shocks. However, evidence on the comparative resilience of SFSCs in relation to conventional supply chains under the pandemic is largely anecdotal [4]. Small chains suffer from a series of intrinsic limitations. These limitations were evident during the pandemic and constitute the main challenges for the future development of SFSCs. Short chains involve small farmers with limited capabilities in terms of access to financing, logistic infrastructure, technology, knowledge, and innovation capacity, and they require specific resources to address these liabilities. Furthermore, SFSCs were severely conditioned by public policies to combat COVID-19.

Notwithstanding the above-mentioned limitations, from a resource-based perspective, SFSCs also present some competitive advantages in a crisis context, such as the current pandemic. Small producers in short chains are often dimensioned according to the available local labor force, so they are not so dependent on immigrant wage workers, and some are able to produce a significant proportion of input resources needed for operation [25]. In addition, the decentralized structure of SFSCs allows the spread of risk among many producers [46]. SFSCs are not capable of replacing the conventional globalized food supply because of their insufficient productive capacity to offer cheap food in volumes that meet demand. However, it is important to note that the dichotomy between long-chain industrialized supply systems and local short chains is not so clear and there is some level of integration in the system. Small producers often sell their produce in both long and short chains, and small operators tend to specialize in certain types of products not offered by large companies. SFSCs integrated into a system that is predominantly based on production-intensive units increase the resilience of the system by serving as a safety net in the event of any type of major disturbance.

### 4.2. Possible Routes to Improve the Sustainability of SFSCs

In the wake of this health crisis, the importance of encouraging local food production and SFSCs to supplement the traditional food supply system and strengthen food supply resilience was reinforced. However, to contribute to the long-term goals of food system resilience, the ability of SFSCs to deliver food in a sustainable way should also be enhanced [46]. Possible routes to improve the sustainability of SFSCs may involve ecological technology and innovation, as well as clear, tailor-made, well-targeted development strategies and policies that support SFSC actors and are aligned with sustainable goals. These aspects are related because policy development can build the basis for sustainable innovation and can help SFSC actors to overcome intrinsic and contextual constraints to innovation adoption.

In terms of technology and innovation, some authors argue that catastrophic global events, such as the COVID-19 pandemic, can trigger complete paradigm shifts, with the introduction of disruptive technologies [75,83]. Notwithstanding the possibility of introducing disruptive technology, sustaining innovation, which entails improving and adapting existing technologies, may be a more pragmatic and short-term solution. Existing technology solutions that are commonly used by long chains can be adapted to fit the low-scale characteristics of SFSCs. Next, we synthetize the main technological and innovation pathways.

Sustainable efficiency-enhancing production and logistics technologies can improve the economic and environmental sustainability of SFSCs. These technologies include e-procurement solutions, intelligent decision support systems and artificial intelligence technology, automation (of production, processing and delivery), outsourced transportation, new farming precise systems that are climate-smart, technologies to reduce carbon emissions and improve input use, biofertilizers, expansion of sustainable micro-agriculture models (e.g., vertical hydroponic farming, rooftop agriculture, as well as office and school gardens), biodynamic cultivation methods, and on-farm production of renewable energy.

Traceability and quality certification technology can contribute to the economic sustainability of SFSCs. These technologies facilitate food traceability, uncover food piracy, and contribute to the social sustainability of SFSCs by safeguarding public health. Traceability and quality certification technology includes technologies based on genomics and bioinformatics to trace and access the genetic authenticity of food products, such as DNA barcoding or DNA genotyping methods. In addition, the digitalization of business activities favors the economic development of SFSCs. This pathway includes the implementation of scale-appropriate innovations, such as biosensor technologies, satellite navigation systems and positioning technologies, the Internet of Things, radio frequency identification technology, electronic food placement, non-cash payment solutions, and big data. Producers in SFSCs can also improve the sustainability and traceability of the system by improving green supplier selection and related methods and approaches [58,83].

Institutional and governance innovation is also important for the sustainable development of SFSCs. Institutional innovation may involve partnerships linking various actors (small local producers, larger companies, academic institutions, nonprofit organizations, government, and consumers), collaborative ventures for innovation, horizontal collaboration involving coopetition arrangements, cooperative organizations to improve access to markets and to gain scale, landbanks and conservation easements, community land trusts, centers of research excellence linked to enterprise and education, or international networks involving farmers.

SFSCs can innovate by exploring new market niches and marketing channels. For example, small farmers can develop new marketing channels by diversifying production through crop rotation or intercropping, enabling harvesting throughout the year, exploring market niches (e.g., organic food and vegan consumers), expanding e-commerce, adopting mobile apps for selling purposes, associating with big e-platforms, as well as complementing agriculture with ecotourism, gastrotourism, and handicrafts. Innovative marketing strategies can facilitate SFSC farmers to better reach consumers, and social marketing studies and programs can help in better understanding consumers' attitudes, perceptions, and the barriers that influence behavior, while also setting the stage for behavior change toward the type of healthy products usually offered by SFSCs.

Regarding public policies to support the sustainable development of SFSCs, the literature emphasizes the importance of incentives for technology adoption and innovation, including financial incentives and knowledge diffusion programs [84,85]. Technology adaptation requires investments and appropriate education for farmers, who are often less experienced in handling such technologies [20].

Another possible government strategy to tackle the economic liabilities of small producers is to provide financial assistance. Financial assistance can assume various forms, including targeted subsidies to support input costs (fuel and other energy sources, transit and parking fees, wholesale market fees, etc.), reimbursement of operational costs, policies that facilitate access to credit (e.g., guaranteed loan programs, interest-free loans, specialized credit services), tax reductions, and tax exemptions.

Governments can also invest in providing operational and support infrastructure. This option refers to investment in market structures and infrastructure, such as market facilities and related infrastructures, warehousing facilities, cooling equipment, refrigerated transport vehicles, and the provision of other assets to decrease operational costs. In addition to tangible assets, governments can assist SFSC producers by offering specific

services, such as providing market- and environment-related studies and information. The provision of market-related information to SFSC actors can support business decisions. The information provided may include data concerning climatic conditions, input and output market prices, information about labor availability, market demand, or information concerning operators' environmental performance.

Related to the COVID-19 pandemic, and future events of this nature, we highlight the importance of lifting commerce barriers, such as the restrictions imposed on open markets, which constituted a real source of discrimination against supermarkets and other big commerce outlets. In addition to alleviating trade restrictions, it is also important to reduce bureaucratic barriers to ensure easier access to markets and policies that enhance equity and fairness within the supply chain by stimulating fair trade practices. The lack of adequate labor during the pandemic also highlights the importance of employment policies, which may involve financial incentives for new employment, providing personal protective equipment, green channels for immigrant workers in the sector, welfare schemes, disease testing and treatment, the redirection of unemployment in the sector, education about the transmission routes, and information about pandemic prevention.

Finally, we emphasize the relevance of governance policies, such as the formation of new institutions for governance, state-supported collaboration between public institutions, SFSC actors, and non-governmental entities toward enhancing sustainability; the creation of public institutions to measure, monitor, and guide sustainable agriculture; and the implementation of certification schemes. SFSCs are very diverse in terms of their organization, structure, and types of actors involved, and policy measures to support these chains need to consider the specificities of each particular chain that is being targeted. However, emerging economies, especially developing countries, are more constrained in their investment options; thus, supranational organizations also have key roles to play, such as by providing liquidity injections or facilitating technology transfer and knowledge acquisition.

## 5. Conclusions

This study provides a review of the status of research concerning SFSC performance during the COVID-19 pandemic from a sustainability perspective. In so doing, this research offers a synthesis that can be influential for policy and practice, while also serving to set the stage for future studies on the topic. The experience of the pandemic presents a momentous opportunity to further explore the topic of the contribution of SFSCs to system sustainability. Notwithstanding the complexity of the analysis, which involves a highly heterogeneous number of realities and ongoing impacts, viewed holistically, this review refutes the general notion that SFSCs were logical contributors to sustainability during the crisis. From an economic perspective, it is possible to conclude that SFSCs in both high- and low-income countries, generally experienced difficulties during the initial stages of the pandemic, mainly resulting from the policies implemented to reduce the spread of the virus, including citizens' confinement, the closure of market outlets, and blocks in imports and movement restrictions that affected input supply, thereby resulting in farmers' loss of income. This also highlights the impact of the institutional dimension of sustainability on the other dimensions of sustainability.

In terms of social well-being, governments' lockdowns increased unemployment and reduced household income in many economies. Nevertheless, SFSCs played a role by supporting the micro-economy of communities and securing local employment during the crisis, while also contributing to food security and providing access to healthy food. The pandemic also served to motivate increased awareness about the hard living conditions of farm workers in some developed economies, thus triggering some political action to tackle this issue. The specific environmental impacts of SFSCs are not addressed in the reviewed literature, a topic that is handicapped by the lack of specific sectorial data, thereby constituting a relevant limitation of this study. Second, the authors acknowledge that the articles reviewed in this study do not contain all pertinent information regarding the effects of the pandemic in each particular market and SFSC system; therefore, the

information is not fully comprehensive. It is also worth noting that most of the reviewed literature consists of conceptual, theoretical, or illustrative articles, with a limited number of quantitative articles being reviewed. In addition, the articles that applied quantitative research methodologies involved non-representative samples of consumers and farmers in their surveys or modeling techniques.

At present, it is still premature to conclude the long-term impacts of the COVID-19 pandemic on the sustainability of SFSCs. However, it is evident that for SFSCs to offer a sustainable alternative and some level of redundancy to globalized production-intensive food chains, in an attempt to increase food system resilience while securing the sustainability of the system, policies and regulations related to food supply and agriculture should tackle the main liabilities of SFSCs; most notably, the possible negative environmental footprint of some short-chain models, the economic vulnerability of SFSC participants, and the lack of efficiency and effectiveness of production and distribution models. First, there is a need to develop more research focused on increasing the resilience of food systems in general, and SFSCs in particular, to disruptive events, such as pandemics and other naturally or unnaturally caused events that have the potential to affect the food chain and the 2030 sustainable agenda of the UN.

Based on this review, other research direction propositions emerge. Considering the significant diversity and heterogeneity of SFSCs, as well as the ongoing status of the pandemic, the exact long-term effects in each market and typology of short chains remain open questions, which can be addressed more systematically when supplementary data become available. According to some of the reviewed literature, and contrary to what public perception and policy discourses suggest, SFSCs do not always represent the most environmentally sustainable option, and more evidence-based insights are needed to validate the sustainability impact of different forms of SFSCs. From the perspective of consumer behavior, future studies can focus on investigating the consequences of the COVID-19 pandemic in terms of consumers' preferences and attitudes toward SFSCs and sustainability issues. Finally, another line of research is related to the long-term impact of the massive increase in online commerce demand during lockdowns. It is important to evaluate the extent to which this trend will persist in the post-pandemic period and how SFSCs can approach the potential opportunities that arise from this significant change in consumer behavior.

**Supplementary Materials:** The research project and corresponding protocol can be downloaded at https://osf.io/dq3rt/?view_only=afca63ea0a604abb901995618192b3ef.

**Author Contributions:** Conceptualization, N.B. and H.A.; methodology, N.B., H.A. and N.M.; software, N.B., H.A. and N.M.; validation, N.B., H.A. and N.M.; formal analysis, N.B., H.A. and N.M.; investigation, N.B., H.A. and N.M.; resources, N.B., H.A. and N.M.; data curation, N.B.; writing—original draft preparation, N.B., H.A. and N.M.; writing—review and editing, N.B., H.A. and N.M.; visualization, N.B., H.A. and N.M.; supervision, N.B.; project administration, N.B.; funding acquisition, N.B. and H.A. All authors have read and agreed to the published version of the manuscript.

**Funding:** This research was funded by IDICA 2020, Project SDGsCONSUM, IPL, NECE-UBI, FCT–Fundação para a Ciência e a Tecnologia, grant number UIDB/04630/2020 and through project Ref. UIDB/04020/2020.

**Institutional Review Board Statement:** Not applicable.

**Informed Consent Statement:** Not applicable.

**Data Availability Statement:** Further data can be accessed from the corresponding authors upon reasonable request.

**Conflicts of Interest:** The authors declare no conflict of interest.

## Appendix A

**Table A1.** Preferred reporting items for Systematic Reviews and Meta-Analyses Extension for Scoping Reviews (PRISMA-ScR) Checklist.

| Section | Item | Prisma-ScR Checklist Item | Reported on Page # |
|---|---|---|---|
| **Title** | | | |
| Title | 1 | Identify the report as a scoping review. | 1 |
| **Abstract** | | | |
| Structured summary | 2 | Provide a structured summary that includes (as applicable): background, objectives, eligibility criteria, sources of evidence, charting methods, results, and conclusions that relate to the review questions and objectives. | n.a. to this journal |
| **Introduction** | | | |
| Rationale | 3 | Describe the rationale for the review in the context of what is already known. Explain why the review questions/objectives lend themselves to a scoping review approach. | 2 |
| Objectives | 4 | Provide an explicit statement of the questions and objectives being addressed with reference to their key elements (e.g., population or participants, concepts, and context) or other relevant key elements used to conceptualize the review questions and/or objectives. | 2 |
| **Methods** | | | |
| Protocol and registration | 5 | Indicate whether a review protocol exists; state if and where it can be accessed (e.g., a web address); and if available, provide registration information, including the registration number. | 3 |
| Eligibility criteria | 6 | Specify characteristics of the sources of evidence used as eligibility criteria (e.g., years considered, language, and publication status), and provide a rationale. | 3 |

**Table A1.** *Cont.*

| Section | Item | Prisma-ScR Checklist Item | Reported on Page # |
|---|---|---|---|
| Information sources | 7 | Describe all information sources in the search (e.g., databases with dates of coverage and contact with authors to identify additional sources), as well as the date the most recent search was executed. | 3 |
| Search | 8 | Present the full electronic search strategy for at least one database, including any limits used, such that it could be repeated. | Protocol |
| Selection of sources of evidence | 9 | State the process for selecting sources of evidence (i.e., screening and eligibility) included in the scoping review. | 3 |
| Data charting process | 10 | Describe the methods of charting data from the included sources of evidence (e.g., calibrated forms or forms that were tested by the team before their use, and whether data charting was done independently or in duplicate) and any processes for obtaining and confirming data from investigators. | 3 |
| Data items | 11 | List and define all variables for which data were sought and any assumptions and simplifications made. | protocol |
| Critical appraisal of individual sources of evidence§ | 12 | If done, provide a rationale for conducting a critical appraisal of included sources of evidence; describe the methods used and how this information was used in any data synthesis (if appropriate). | n.a. |
| Synthesis of results | 13 | Describe the methods of handling and summarizing the data that were charted. | 3 |
| **Results** | | | |
| Selection of sources of evidence | 14 | Give numbers of sources of evidence screened, assessed for eligibility, and included in the review, with reasons for exclusions at each stage, ideally using a flow diagram. | 3 |
| Characteristics of sources of evidence | 15 | For each source of evidence, present characteristics for which data were charted and provide the citations. | Protocol |

**Table A1.** *Cont.*

| Section | Item | Prisma-ScR Checklist Item | Reported on Page # |
|---|---|---|---|
| Critical appraisal within sources of evidence | 16 | If done, present data on critical appraisal of included sources of evidence (see item 12). | n.a. |
| Results of individual sources of evidence | 17 | For each included source of evidence, present the relevant data that were charted that relate to the review questions and objectives. | Protocol |
| Synthesis of results | 18 | Summarize and/or present the charting results as they relate to the review questions and objectives. | n.a. |
| **Discussion** | | | |
| Summary of evidence | 19 | Summarize the main results (including an overview of concepts, themes, and types of evidence available), link to the review questions and objectives, and consider the relevance to key groups. | 5, 6 |
| Limitations | 20 | Discuss the limitations of the scoping review process. | 16 |
| Conclusions | 21 | Provide a general interpretation of the results with respect to the review questions and objectives, as well as potential implications and/or next steps. | 16 |
| **Funding** | | | |
| Funding | 22 | Describe sources of funding for the included sources of evidence, as well as sources of funding for the scoping review. Describe the role of the funders of the scoping review. | 17 |

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
