# Peer review of "Scoping Challenges and Opportunities Presented by COVID-19 for the Development of Sustainable Short Food Supply Chains"

_sustainability, doi:10.3390/su142114475_

Round 1
Reviewer 1 Report
Major comments: In a review article, instead of long explanations of the findings, it is better to present the main findings in tables, and further explanations should come briefly in the text so as not to be boring for the reader.
Minor comments
Introduction: The last paragraph of the Introduction can be deleted.
Background is too long and can be integrated in Introduction.
Results: The explanation of the findings is too long. It is better to present a table of the characteristics of the included studies in the scoping review by combining two tables 1 and 2.
Some columns e.g. Journal doesn't seem very necessary in the table.
Discussion: Some findings are presented in Discussion:
“Table 2 summarizes the effects of the pandemic in some selected economies based on the reviewed literature”.
Author Response
First, we would like to express our gratitude to reviewer 1 for his constructive comments and for the time he generously spent in the analysis of our work. We consider his comments to be very pertinent and we believe that we have improved our article based on his suggestions. We would like to advance that we agreed with most of the recommendations made by reviewer 1, and decided to act accordingly by implementing the necessary changes in our manuscript. Next, we individually address all the suggestions made by reviewer 1.
Reviewer 1 - Major comments: In a review article, instead of long explanations of the findings, it is better to present the main findings in tables, and further explanations should come briefly in the text so as not to be boring for the reader.
R: We synthetized the content of the article and inserted some sub-titles in the discussion section, to make it more readable and “less boring”. Table 2, which was revised, also synthesizes some of the main findings.
Reviewer 1: Introduction: The last paragraph of the Introduction can be deleted.
R: The last paragraph of the introduction was deleted as suggested.
Reviewer 1: Background is too long and can be integrated in Introduction.
R: Considering this comment from the reviewer, we decided to reduce the length of the background section and integrate part of its previous content in the introduction, that was renamed as “1. Introduction and background”.
Reviewer 1: Results: The explanation of the findings is too long. It is better to present a table of the characteristics of the included studies in the scoping review by combining two tables 1 and 2. Some columns e.g. Journal doesn't seem very necessary in the table.
R: The thank the reviewer for this suggestion. We decided to simplify table 2. However, it was not practical to integrate the two tables, since they provide very distinct data. We also opted for not deleting the column mentioned by the reviewer because it highlights the importance of the Sustainability journal.
Reviewer 1: Discussion: Some findings are presented in Discussion:
“Table 2 summarizes the effects of the pandemic in some selected economies based on the reviewed literature”.
R: Table 2 was simplified and moved to the results section, as suggested by the reviewer. We thank the reviewer for this well thought suggestion. Makes perfect sense.
Reviewer 2 Report
The paper is interesting and well written. I propose acceptance after minor revisions related to:
1) a revision of the english language in order to improve readability and eliminating typos;
2) Improving Table 2 style: the table is very large and divided in different pages, it is very hard to read it. I think that reducing the table will be very useful;
3) the discussion can be divided in different subèaragraph connected with the subsection of the results (social, institutional and environemental sustainability). A parallelism is very welcomed within the discussion exposure.
Author Response
We would like to express our gratitude to reviewer 2 for his constructive comments and for the time he generously spent in the analysis of our work. We consider his comments to be very pertinent and we believe that we have improved our article based on his suggestions. We would like to advance that we agreed with most of the recommendations made by reviewer 2, and decided to act accordingly by implementing the necessary changes in our manuscript. Next, we individually address all the suggestions made by reviewer 2.
Reviewer 2 - A revision of the english language in order to improve readability and eliminating typos;
R: The English of the article was revised. We recurred to an English language editing service from Elsevier that involves English native reviewers.
Reviewer 2 - Improving Table 2 style: the table is very large and divided in different pages, it is very hard to read it. I think that reducing the table will be very useful;
R: We decided to simplify table 2 and resume its content to make it more readable, as suggested by reviewer 2.
Reviewer 2 - The discussion can be divided in different subèaragraph connected with the subsection of the results (social, institutional and environemental sustainability). A parallelism is very welcomed within the discussion exposure.
We thank the reviewer for this suggestion. We inserted some sub-chapters to facilitate the reading and interpretation of the discussion section and make the contribution of the article more clear. In some sense, a parallelism was established when discussing the possible routes to improve the sustainability of SFSCs in sub-chapter “5.2. Possible routes to improve the sustainability of SFSCs.”
Reviewer 3 Report
1. The section of abstract need to re.write in more details.
2. Please fix the typos in the English language, and correct all grammatical and syntax errors.
3. It will be useful in the conclusion section, to mention some of the future directions.
4. Need to check the overall format of the paper.
5. Authors should clearly structure the manuscript to clearly highlight its contribution.
6. Please explain more about method, including advantage and disadvantages.
7. Please cite following papers as they are related to your study:
8. Recent papers should be added to enrich the outputs.
Author Response
First, we would like to express our gratitude to reviewer 3 for his constructive comments and for the time he generously spent in the analysis of our work. We consider his comments to be very pertinent and we believe that we have improved our article based on his suggestions. We would like to advance that we agreed with most of the recommendations made by reviewer 3, and decided, when possible, to act accordingly by implementing the necessary changes in our manuscript. Next, we individually address all the suggestions made by reviewer 3.
Reviewer 3 - 1. The section of abstract need to re.write in more details.
We thank reviewer 3 for this suggestion. However, according to the instructions of the Sustainability journal, the abstract should be a total of 200 words maximum. The current abstract is very synthetic to comply with this requisite, its length is 198 words. So, there is not space to add more detail. Nevertheless, we think that the current abstract covers all main conclusions of the article.
Reviewer 3 - Please fix the typos in the English language, and correct all grammatical and syntax errors.
R: The English of the article was revised. We recurred to an English language editing service from Elsevier that involves English native reviewers.
Reviewer 3 - Need to check the overall format of the paper.
R: The overall format of the article was revised. We believe it is now in the desired format.
Reviewer 3 - Authors should clearly structure the manuscript to clearly highlight its contribution.
R: R: We thank the reviewer for this suggestion. We inserted some sub-chapters to facilitate the reading and interpretation of the discussion section and make the contribution of the article more clear. In addition, the contribution of the paper was highlighted in the conclusions sector in the next paragraph:
“This study provides a review of the status of research concerning SFSC performance during the COVID-19 pandemic from a sustainability perspective. In so doing, this research offers a synthesis that can be influential for policy and practice, while also serving to set the stage for future studies on the topic.”
Reviewer 3 - Please explain more about method, including advantage and disadvantages.
R: In the Materials and Methods section of the paper we explain the advantages of the method employed (Scoping Review) comparing its advantages in relation to a systematic literature review. The comparison was made in the following paragraph:
As highlighted by Tricco et al. (2016), scoping reviews differ from systematic literature reviews in the sense that while the former is exploratory in nature, being commonly used to examine new areas that are emerging, the latter is more suitable for addressing specific questions related to more mature research topics. Considering the exploratory nature of the research, the authors opted to conduct a scoping review instead of a systematic review.”
We also add the folowing explanation in the revised manuscript:
Although conducted for different purposes compared to systematic reviews, scoping reviews also require rigorous procedures to ensure that the results are trustworthy [2].
Reviewer 3 - Please cite following papers as they are related to your study:
Hashemzahi, P., Azadnia, A., Galankashi, M. R., Helmi, S. A., & Rafiei, F. M. (2020). Green supplier selection and order allocation: a nonlinear stochastic model. International Journal of Value Chain Management, 11(2), 111-138.
Galankashi, M. R., Chegeni, A., Soleimanynanadegany, A., Memari, A., Anjomshoae, A., Helmi, S. A., & Dargi, A. (2015). Prioritizing green supplier selection criteria using fuzzy analytical network process. Procedia Cirp, 26, 689-694.
R: We thank the reviewer for this suggestion. The papers are now cited in the revised version of the manuscript, when discussing the importance of improving the methods for supplier selection - section “5.2. Possible routes to improve the sustainability of SFSCs”.
Round 2
Reviewer 1 Report
Most of the comments were considered and improvements were made in the manuscript.